# Nanoparticles Assembled CdIn_2_O_4_ Spheres with High Sensing Properties towards n-Butanol

**DOI:** 10.3390/nano9121714

**Published:** 2019-12-01

**Authors:** Weiping Liu, Ximing Zhang, Zhaofeng Wang, Ruijian Wang, Chen Chen, Chengjun Dong

**Affiliations:** 1College of Instrumentation & Electrical Engineering, Key Laboratory of Geophysical Exploration Equipment, Ministry of Education of China, Jilin University, Changchun 130026, Jilin, China; liuweiping2014@jlu.edu.cn (W.L.); xmzhang18@mails.jlu.edu.cn (X.Z.); wzf19@mails.jlu.edu.cn (Z.W.); wangrj18@mails.jlu.edu.cn (R.W.); 2School of Materials Science and Engineering, Yunnan University, Kunming 650091, Yunnan, China

**Keywords:** gas sensor, n-butnaol, CdIn_2_O_4_, spheres, solvothermal method

## Abstract

Cd/In-glycerate spheres are synthesized through a simple solvothermal method. After thermal treatment, these Cd/In-glycerates can be converted into CdIn_2_O_4_ spheres. Many characterization methods were performed to reveal the microstructure and morphology of the CdIn_2_O_4_. It was found that pure CdIn_2_O_4_ phase was obtained for the Cd/In starting materials at ratios of 1:1.6. The CdIn_2_O_4_ spheres are composed by a large number of nanoparticles subunits. The CdIn_2_O_4_ sphere-based sensor exhibited a low detection limit (1 ppm), high response (81.20 to 500 ppm n-butanol), fast response (4 s) and recovery (10 s) time, good selectivity, excellent repeatability, and stability at 280 °C. Our findings highlight the possibility to develop a novel gas sensor based on CdIn_2_O_4_ for application in n-butanol detection with high performance.

## 1. Introduction

With the rapid development of society and progress in industry, the increasing air pollution is becoming an urgent global problem [1,2]. In particular, volatile organic compounds (VOCs) are harmful for both human health and environmental safety, as their concentration exceeds a critical threshold [3,4]. As a typical kind of VOC, n-butanol (CH_3_(CH_2_)_3_OH) is an excellent solvent, organic synthesis raw material, and intermediate as well as an extracting agent, and is widely used in laboratories and industry [5]. A series of serious symptoms, such as dizziness, headache, somnolence, and dermatitis, etc., could be caused by an n-butanol atmosphere [6]. Therefore, the threshold limit value of n-butanol in the workplace has been specified to be below 200 mg/m^3^ (e.g., 200 mg/m^3^ and 152 mg/m^3^ for the United States) [7]. Moreover, explosion or fire may occur if 1.45–11.25% n-butanol is mixed in air when the temperature is higher than its flash point of 35 °C. Consequently, it is highly necessary to effectively detect n-butanol in laboratory and factory settings with fast speed and excellent selectivity. For now, several monitoring methods have been reported for n-butanol detection, in which metal oxide semiconductors such as SnO_2_ [8], ZnO [9], WO_3_ [10], Fe_2_O_3_ [6], and their hybrid-based gas sensors [11,12,13] have drawn considerable attention because of their advantages of sensing properties and low cost. Yet, it is still necessary to develop an n-butanol gas sensor based on novel sensing materials.

As is well known, CdIn_2_O_4_ is a typical n-type semiconductor, which is widely applied for materials of transparent conducting oxides (TCO), with a bandgap of 3.2 eV [14]. Studies on CdIn_2_O_4_ exhibited gas sensing performance to formaldehyde [15,16], ethanol [14,17,18], Cl_2_ [19], acetone [20], petroleum gas [21], and H_2_S [22] in nanoparticles, thin films, nanofibers, octahedrons, and so on. However, a great challenge must be overcome in order to synthesize a pure phase of CdIn_2_O_4_ with controllable morphology using a simple, low-cost method. Metal alkoxides appear to be a novel precursor-directed synthetic strategy to fabricate inorganic nanomaterials, in which the metal provides extra electrons to coordinate with the hydroxyl groups of alcohols to form stable compounds in solids [23]. Here, we developed a facile solvothermal method followed by a simple thermal treatment to synthesize CdIn_2_O_4_ spheres composed of nanoparticles using a novel precursor-mediate. The CdIn_2_O_4_ spheres were further applied as sensing materials for n-butanol detection, which exhibited excellent sensing performance. Moreover, the gas sensing mechanism for the CdIn_2_O_4_ based sensor has been discussed from the aspect of the surface chemical state.

## 2. Materials and Methods

### 2.1. Synthesis of CdIn_2_O_4_ Spheres

All of the chemical reagents including cadmium nitrate (Cd(NO_3_)_2_·4H_2_O), indium nitrate (In(NO)_3_·xH_2_O), glycerol, and isopropanol were purchased from Aladdin Industrial, Inc. (Shanghai, China). All these chemicals were utilized as received without further purification. In addition, deionized water was involved in all experiments. In a typical synthesis of Cd-In precursor, 1 mmol Cd(NO_3_)_2_·4H_2_O and different amounts of In(NO)_3_·xH_2_O (2 mmol, 1.6 mmol, and 1.2 mmol) were dissolved into the mixture of 10 mL glycerol and 50 mL isopropanol under vigorous stirring for 1 h. Then, the resulting mixture was transferred into a 100 mL Teflon autoclave (Yusheng Instrument, Shanghai, China), which was treated at 180 °C for 12 h. When it was cooling down, the white product (i.e., Cd-In precursor) was centrifuged several times with deionized water and ethanol. Then the powder was collected after drying at 60 °C in air. Finally, the yellow product was obtained by annealing the dried Cd-In precursor (grayish white in color) with a heating rate of 5 °C/min to 800 °C and kept at 800 °C for 1 h in air.

### 2.2. Materials Characterization

The crystal phase was examined by X-ray diffraction (XRD, Rigaku TTRIII, Rigaku Corporation, Tokyo, Japan). Cu K_α_ radiation (1.54056 Å) was used. The microstructure of CdIn_2_O_4_ was investigated with a field-emission scanning electron microscope (FESEM, FEI QUANTA 200, FEI Company, Hillsboro, OR, USA) and transmission electron microscopy (TEM, JEM-2100, JEOL, Tokyo, Japan). Thermogravimetric analysis (TGA) was performed in a flow of air from room temperature to 800 °C, taking a heating rate of 10 °C/min. The N_2_ adsorption-desorption measurement was conducted at 77 K by a Micromeritics ASAP 2010 system (Micromeritics, Norcross, GA, USA) after degassing at 300 °C for 2 h. The chemical states of CdIn_2_O_4_ was recorded using X-ray photoelectron spectroscopy (XPS) (Thermo Fisher Scientific Co., Ltd., Waltham, MA, USA).

### 2.3. Sensor Fabrication and Testing

A slurry was formed by mixed the as-prepared CdIn_2_O_4_ with suitable deionized water. Then, it was coated onto a ceramic tube with 1 mm and 4 mm outer diameter and length, respectively. Two circle Au electrodes were separately pasted onto the two ends of the ceramic tube, which was further connected with four Pt wires. After being dried at 120 °C for 2 h, a thermal treatment was carried out in air 400 °C for 1 h to make good contact between the Au electrodes and sensing materials. After installing a Ni-Cr alloy wire into the tube, all wires were connected to a base to perform testing in a WS-30A system (Weisheng Instruments Co., Zhengzhou, China). CdIn_2_O_4_ is an n-type semiconductor, thus the sensor response can be calculated by the change in sensor resistance (*R*_a_/*R*_g_) in air (*R*_a_) and target gas (*R*_g_). The response time and recovery time are generally defined as the time for 90% of the initial equilibrium resistance change.

## 3. Results

Previous reports have shown that the molar ratio of Cd/In in the initial materials could significantly affect the product purity [14,19,24]. Therefore, the typical Cd/In molar ratios of 1:2, 1:1.6, and 1:1.2 were studied based on our experimental strategy, which were characterized by XRD measurement, as displayed in Figure 1. It is evidenced that the observed diffraction peaks of the product can be well indexed to the cubic CdIn_2_O_4_ phase (JCPDS: 70-1680) for Cd:In = 1:1.6 in Figure 1b. However, the byproducts of In_2_O_3_ (JCPDS: 71-2195) appeared along with CdIn_2_O_4_ phase with a stoichiometric ratio of Cd:In = 1:1.2 (Figure 1a), due to the different coordinated capacity of glycerol with Cd^2+^ and In^3+^ ions. Meanwhile, CdO phase (JCPDS: 75-0592) could be detected when the Cd/In ratio was reduced to 1:1.2 of the starting materials (Figure 1c). Thus, the Cd/In molar ratio of 1:1.6 was suitable to synthesize pure CdIn_2_O_4_ using a solvothermal method followed by thermal treatment using glycerol and isopropanol as the mixed solvent, which will be mainly studied in the following.

Figure 2 shows the morphologies of the Cd/In precursor and the as-formed pure CdIn_2_O_4_ spheres. The Cd/In precursor exhibits connected spheres with loose solids in nature (Figure 2a,b). After thermal treatment under air atmosphere, the CdIn_2_O_4_ is preserved with a connected structure of the Cd/In precursor, as shown in Figure 2c. The magnified observations in Figure 2d further reveal the CdIn_2_O_4_ sphere is composed of a lot of nanoparticles subunits. The nanoparticles are determined to be dozens to more than one hundred nanometers from the broken CdIn_2_O_4_ sphere, as depicted in Figure 3c. As displayed by TEM investigations in Figure 3a,b, the random cavities are elucidated by the clear contrast. A typical individual CdIn_2_O_4_ sphere has a diameter of about 650 nm (Figure 3b). In addition, the visible lattice spacing is about 0.28 nm from the High Resolution Transmission Electron Microscope (HRTEM) image (Figure 3d), corresponding to the distance between (311) crystal planes of cubic CdIn_2_O_4_. The Energy Dispersive X-Ray Spectroscopy (EDS) analysis in Figure 4a shows the existence of Cd (27.60 wt.%), In (56.64 wt.%) and O (15.77 wt.%), which is close to the stoichiometric quantities in CdIn_2_O_4_. At the same time, the EDS mapping reveals the homogeneous distribution of Cd, In, and O elements (Figure 4c–e) throughout a typical CdIn_2_O_4_ sphere (Figure 4b). As determined by N_2_ adsorption-desorption measurement, these CdIn_2_O_4_ spheres posse a Brunauer–Emmett–Teller (BET) specific surface area of 10.68 m^2^/g with the pore sizes mostly around 30 nm (Figure 5). The relatively small specific surface area can be explained by the large accumulation of CdIn_2_O_4_ nanoparticles.

As a polyalcohol, glycerol contains three hydroxyl groups, which have a strong propensity to coordinate with metal ions to produce a stable structure [23]. Thus, the growth mechanism of CdIn_2_O_4_ spheres could be explained as follows. First, by coordinating with Cd and In ions, Cd/In-glycerate spheres were formed. Isopropanol played an important role in the formation of Cd/In-glycerate. In isopropanol, the metal ions are easily dissolved. Moreover, isopropanol can significantly reduce the viscosity of the reaction solution, resulting in the increase of the reaction opportunity between both Cd^2+^ and In^3+^ ions and glycerol. After thermal treatment at 800 °C, Cd/In-glycerate spheres were converted into nanoparticles and assembled CdIn_2_O_4_ spheres.

Figure 6 shows the TGA curve for the formation of CdIn_2_O_4_ in air with a heating rate of 10 °C/min. At the lower temperature ranges from room temperature to 225 °C, the Cd/In-glycerate exhibits 11.84% weight loss, which is attributed to the adsorbed or entrapped organic solvent, as well as water. A large weight loss of 14.01% occurs when the temperature further increases to 365 °C. Meanwhile, an exothermic effect can be noticed in the Differential Scanning Calorimeter (DSC) curve at 280.69 °C, suggesting the decomposition of the organic parts of the Cd/In-glycerate precursor [23,25]. Subsequently, a small loss of 3% in weight is observed at higher temperatures up to 600 °C, which is ascribed to the crystal of CdIn_2_O_4_. Thus, 28.85% weight loss is calculated in total after thermal treatment, due to the release of CO_2_ and water vapor, leading to the conversion of smooth Cd/In-glycerate to nanoparticles assembled in CdIn_2_O_4_ spheres.

Based on the above structural features of CdIn_2_O_4_, it is expected to serve as a good gas-sensing material. Thus, the sensor device was fabricated using CdIn_2_O_4_ spheres as sensing materials, and n-butanol was selected as the detecting molecule. Because the operating temperature is a basic parameter for a gas sensor to determine the sensing performance, the gas responses to 500 ppm n-butanol were examined first (Figure 7a). Clearly, the response to n-butanol gas initially increases below 280 °C. Subsequently, it decreases when the temperature is further increased. With regard to the low temperature, the reaction between absorbed oxygen species on the CdIn_2_O_4_ surface and the gas molecules is not completely activated. However, the insufficient adsorption of gas molecules will reduce the response over high temperatures. Clearly, the maximum response of 81.20 is observed at 280 °C, which is considered the optimum temperature to investigate sensing-related properties in order to evaluate the gas sensor.

Figure 7b shows the dynamic response characteristics of a CdIn_2_O_4_ sphere-based gas sensor to various n-butanol concentrations ranging from 1 to 500 ppm, illustrating an increasing tendency with the tested gas concentrations. Specifically, the corresponding responses are 1.48, 6.35, 10.46, 18.92, 27.32, 32.57, 40.72, 62.54, and 81.20 to 1, 5, 10, 30, 50, 70, 100, 300, and 500 ppm n-butanol, respectively, as depicted in Figure 7c. Importantly, the detection limit of the sensor could reach 1 ppm n-butanol, as magnified in the inset of Figure 7b. As the n-butanol concentrations increase from 1 to 100 ppm, a big increase in response is observed. However, the gas response value gradually increases with respect to higher n-butanol concentrations, because the surface of sensing materials is largely covered by the gas molecules. Moreover, the fast response and recovery times of the sensor to n-butanol are found to be 4 s and 10 s (Figure 7d). On top of this, an excellent reversible response-recovery switch can be seen from the five reversible cycles of the sensor, based on CdIn_2_O_4_ spheres of up to 300 ppm n-butanol at 280 °C (Figure 7e). Taking the n-butanol concentrations of 100 ppm and 300 ppm as examples, long-term stability has been investigated. From Figure 7f, a slight fluctuation is observed for 30 days, suggesting excellent performance in stability.

However, selectivity is also an important parameter in evaluating the comprehensive sensing properties. Therefore, the selectivity of our sensor was examined against other interference gases by measuring the response to 500 ppm acetone, methanol, isopropanol, formaldehyde, toluene, xylene, benzene, ammonia, and ethanol at optimum operating temperature, as shown in Figure 8. Clearly, the response to n-butanol is much higher than the interference gases, indicating a good selectivity. The good selectivity could be explained by the fact that the operating temperature of 280 °C provides sufficient energy to favor n-butanol gas molecule adsorbtion and desorbtion and react on the CdIn_2_O_4_ surface. At lower temperatures, insufficient energy is not in favor of reactions between gas molecules and absorbed oxygen species. However, the absorbed gas molecules could greatly decrease at relatively higher temperatures.

Taken together, the CdIn_2_O_4_ sphere-based gas sensor exhibits excellent gas-sensing properties towards n-butanol detection with respect to low detection limit, wide detecting range, high response, fast response and recovery times, as well as good stability. The pioneering work by Seiyama et al. revealed that the process of adsorption and desorption of VOC molecules produced a distinct change in electrical conductivity of semiconductors, which is the basic principle for gas sensors [26]. Thus, the gas sensitive mechanism was explored, as shown schematically in Figure 9d–f. As one of typical n-type semiconductor, the adsorbed oxygen molecules on the surface of CdIn_2_O_4_ will capture electrons from the conduction band to form active oxygen species (such as O^2−^, O^−^, and O_2_^−^) [27], leading to an increase in the resistance (Figure 9e). When the CdIn_2_O_4_-based sensor is exposed to an n-butanol atmosphere, these oxygen species will react with n-butanol molecules, hence, the resistance will decrease by releasing the captured electrons back (Figure 9f). The variation in resistance is strongly dependent on the n-butanol concentrations. The chemical state of CdIn_2_O_4_ spheres were studied in detail by XPS measurement, as shown in Figure 9a–c, which confirms the component elements Cd, In, and O of the CdIn_2_O_4_ spheres. The XPS spectrum of Cd 3d shows that the two strong peaks at 405.57 eV and 412.33 eV are related to Cd 3d_5/2_ and Cd 3d_3/2_ in Figure 9a, respectively, demonstrating the Cd^2+^ of CdIn_2_O_4_ [28,29]. Meanwhile, two characteristic peaks in the In 3d spectrum located at the binding energy of 444.73 eV and 452.29 eV are assigned to In 3d_5/2_ and In 3d_3/2_ in Figure 9b, corresponding to the valence of In^3+^. Significantly, the adsorbed oxygen species are confirmed by the peak at the higher binding energy of 532.20 eV, apart from the lattice oxygen with a binding energy of 530.12 eV in the XPS spectrum of O 1 s (Figure 9c), which is reactive with the n-butanol molecules [30]. The plausible reactions are given in the following [31,32]:(1)$$O_{2}\left(\mathrm{gas}\right)\to O_{2}\left(\mathrm{adsorbed}\right)$$
(2)$$O_{2}{(\mathrm{adsorbed})\;+\;e}^{-}\to O_{2}^{-}$$
(3)$$O_{2}^{-}{\;+\;e}^{-}\to 2O^{-}$$
(4)$$O^{-}{\;+\;e}^{-}\to O^{2-}$$
(5)$${\mathrm{CH}}_{3}{{(\mathrm{CH}}_{2})}_{3}\mathrm{OH}\;+\;12O^{-}\to 4{\mathrm{CO}}_{2}{\;+\;5H}_{2}{O\;+\;12e}^{-}$$
(6)$${\mathrm{CH}}_{3}{{(\mathrm{CH}}_{2})}_{3}\mathrm{OH}\;+\;12O^{2-}\to 4{\mathrm{CO}}_{2}{\;+\;5H}_{2}{O\;+\;24e}^{-}$$

In comparison to reports with respect to n-butanol gas sensors, as listed in Table 1, our CdIn_2_O_4_-based sensor exhibits comparable performance in response and recovery times and limit of detection. The response of CdIn_2_O_4_ towards n-butanol could be further enhanced by functionalizing noble nanoparticles (Pt, Pd, Au, and Ag) or their oxides (PdO and PtO_2_) [33]. In addition, the CdIn_2_O_4_ spheres could be a backbone to couple with other sensing materials such as SnO_2_, ZnO, In_2_O_3_, Fe_2_O_3_ etc., to create heterojunctions, thus the sensing properties can be greatly improved [34,35]. Moreover, suitable additives or templates (hard templates, soft templates, and bio-templates) may be introduced into the reacting system to tune the morphologies to increase the specific surface area of CdIn_2_O_4_ [36,37], which endows more reactive sites to favor the enhancement of sensing performance.

## 4. Conclusions

In summary, CdIn_2_O_4_ spheres were solvothermally synthesized using glycerol as coordinated agents. The effect of Cd/In molar ratios has been explored. The CdIn_2_O_4_ sphere-based gas sensor displays a low detection limit of 1 ppm with a response value of 1.48 and wide detecting range from 1 to 500 ppm at an operating temperature of 280 °C. Moreover, high selectivity against acetone, methanol, isopropanol, formaldehyde, toluene, xylene, benzene, ammonia, and ethanol is investigated. These results demonstrate that CdIn_2_O_4_ spheres represent a promising sensing material in detecting n-butanol gas.

## Figures and Tables

**Figure 1 nanomaterials-09-01714-f001:**
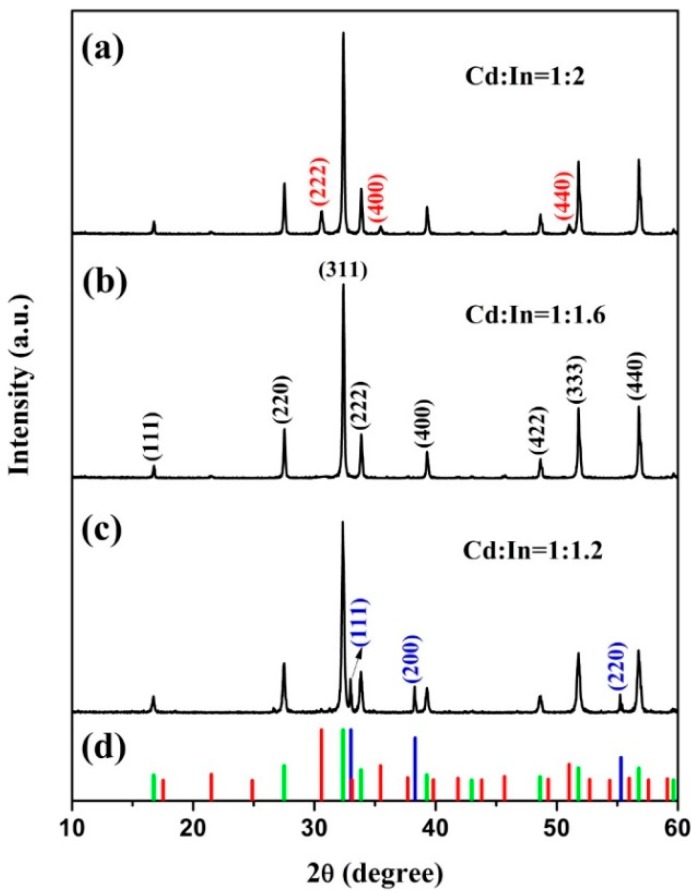
XRD patterns of the CdIn_2_O_4_ with typical Cd/In molar ratios of (**a**) Cd:In = 1:2, (**b**) Cd:In = 1:1.6, (**c**) Cd:In = 1:1.2, and (**d**) the vertical line represents JCPDS of In_2_O_3_ (71-2195, Red), CdO (75-0592, Blue), and CdIn_2_O_4_ (70-1680, Green).

**Figure 2 nanomaterials-09-01714-f002:**
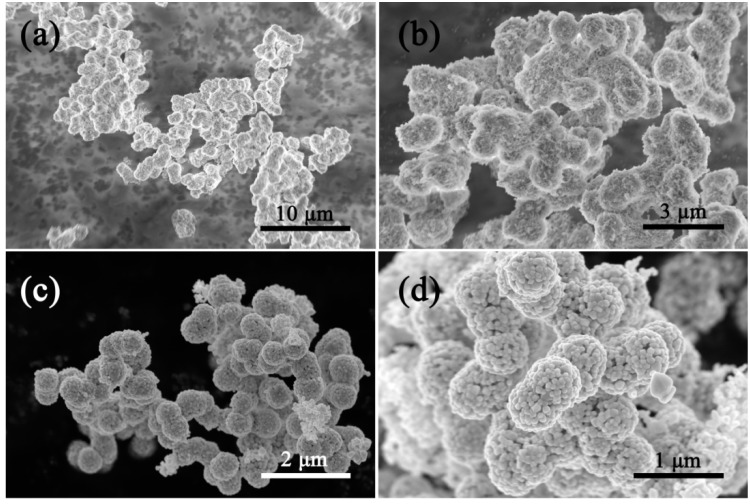
FESEM images of (**a**,**b**) Cd/In precursor (1:1.6), and (**c**,**d**) CdIn_2_O_4_.

**Figure 3 nanomaterials-09-01714-f003:**
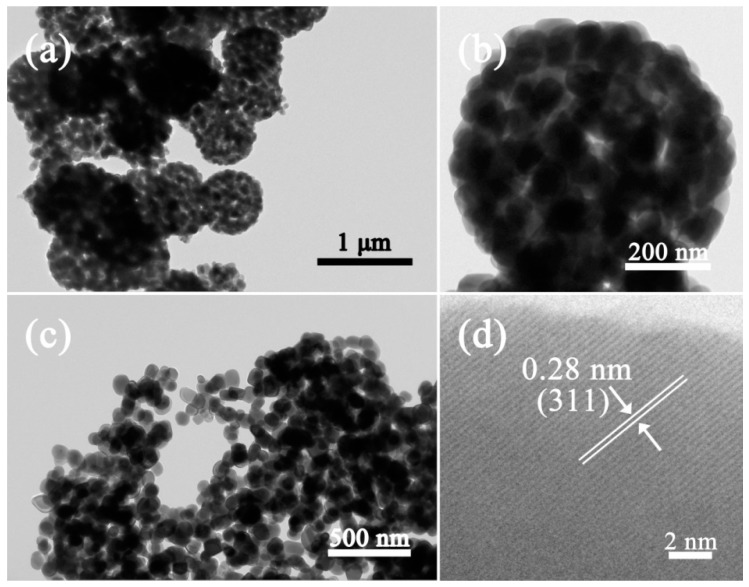
TEM images of (**a**) overview, (**b**) typical, and (**c**) broken CdIn_2_O_4_ spheres, and (**d**) the corresponding HRTEM image.

**Figure 4 nanomaterials-09-01714-f004:**
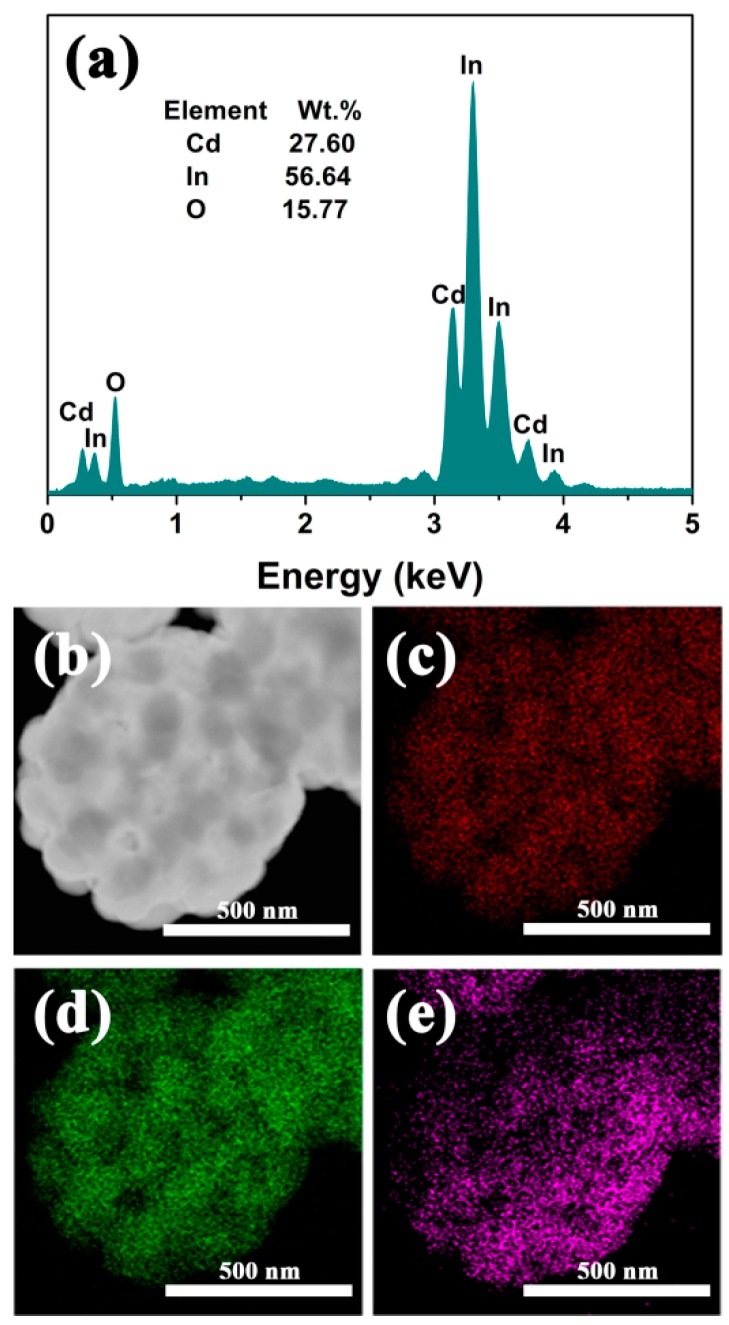
EDS analysis of CdIn_2_O_4_ spheres (**a**), and the elemental distribution mapping of Cd (**c**), In (**d**), and O (**e**) for a selected CdIn_2_O_4_ sphere (**b**).

**Figure 5 nanomaterials-09-01714-f005:**
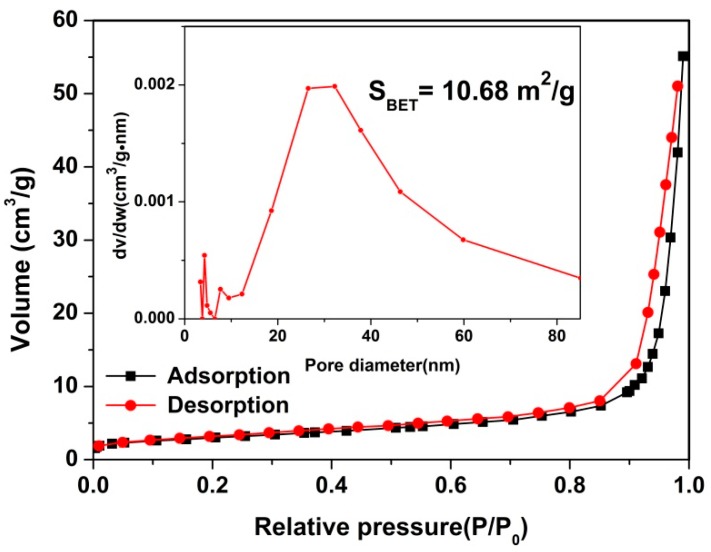
N_2_ adsorption-desorption isotherm of CdIn_2_O_4_ (inset shows the pore size distributions calculated by Barrett-Joyner-Halenda (BJH) method).

**Figure 6 nanomaterials-09-01714-f006:**
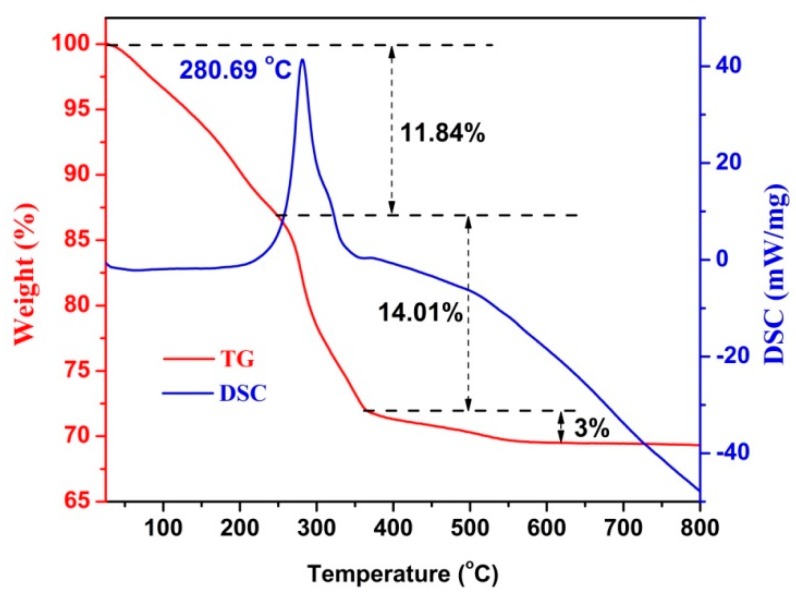
TGA-DSC curves for the formation of CdIn_2_O_4_.

**Figure 7 nanomaterials-09-01714-f007:**
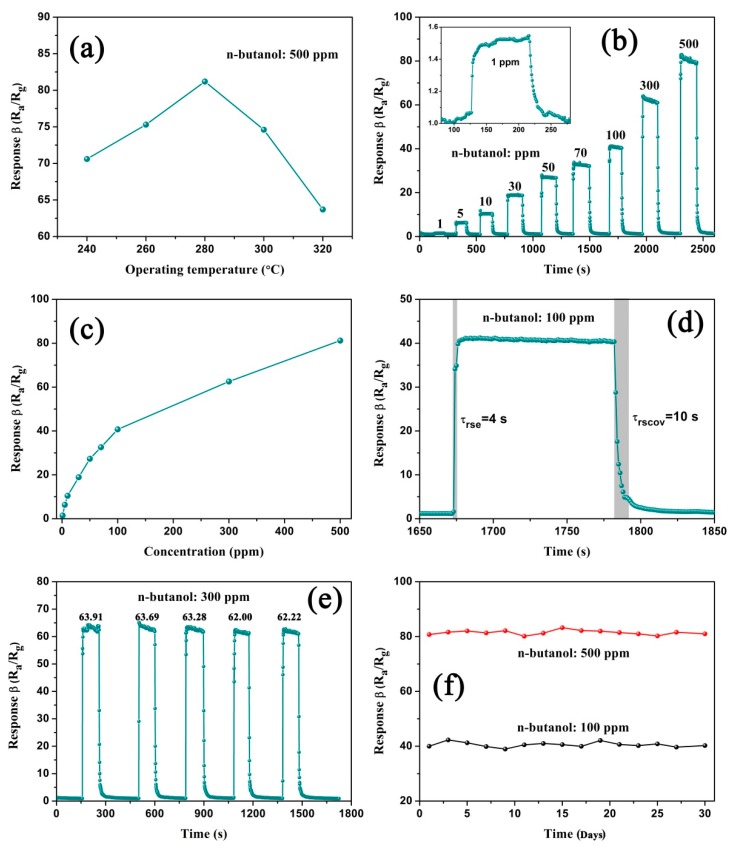
(**a**) The gas response of the CdIn_2_O_4_ sphere-based sensor of up to 500 ppm of n-butanol as a function of operating temperature; (**b**) Dynamic sensing characteristics of the sensor to n-butanol at 280 °C; (**c**) The relationship between calculated responses and n-butanol concentration. (**d**) The response and recovery times for 100 ppm n-butanol; (**e**) The freproducibility of the sensor up to 500 ppm n-butanol; (**f**) Long-term stability of the sensor up to 100 ppm and 500 ppm n-butanol for 30 days.

**Figure 8 nanomaterials-09-01714-f008:**
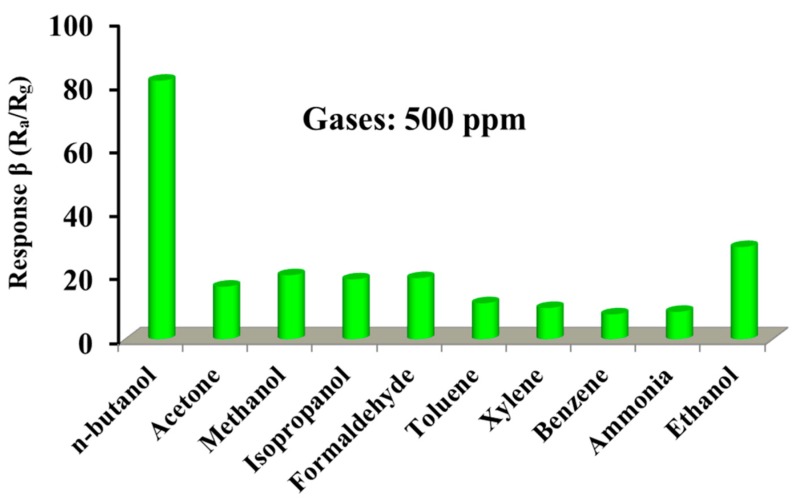
Selectivity of the CdIn_2_O_4_ sensor towards different interfering gases with 500 ppm concentration at 280 °C.

**Figure 9 nanomaterials-09-01714-f009:**
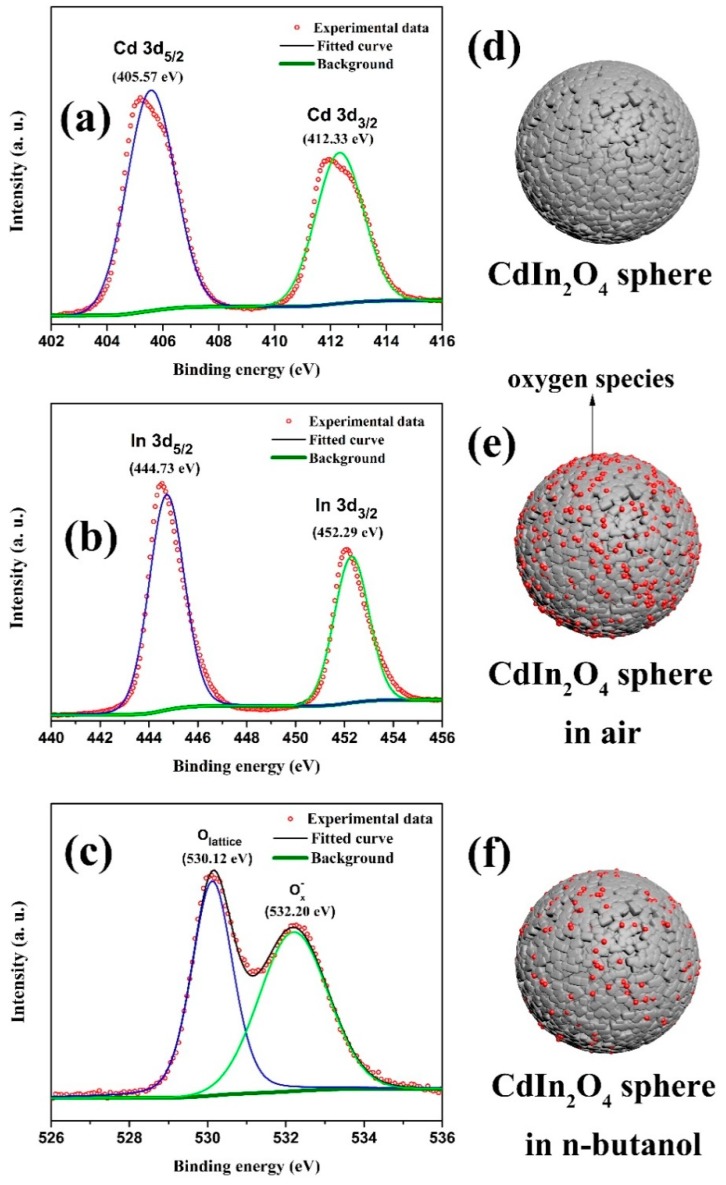
The XPS spectra of (**a**) Cd 3d, (**b**) In 3d, and (**c**) O 1s, and the schematic illustration of CdIn_2_O_4_ (**d**) sphere, (**e**) in air, and (**f**) in n-butanol.

**Table 1 nanomaterials-09-01714-t001:** Comparison of n-butanol sensing performance based on different nanomaterials.

Materials	Microstructures	Concentration (ppm)	T (°C)	Response	Response and Recovery Times (s)	Limit of Detection (ppm)	Ref.
SnO_2_	Hollow cubes	100	310	75.7	2.1/17 (20 ppm)	1	[8]
ZnO	Hollow spheres	100	385	57.6	23/13 (100 ppm)	10	[9]
α-Fe_2_O_3_	Shuttle-shaped	100	250	145	20/55 (50 ppm)	10	[6]
In_2_O_3_	Nanoparticles	50	140	97	45/65 (10 ppm)	5	[38]
CuO	Micro-sheets	1000	160	69.73	--	10	[5]
Co_3_O_4_	Porous	100	100	21	146/90 (100 ppm)	<5	[39]
CdIn_2_O_4_	Spheres	500	280	81.20	4/10 (100 ppm)	1	This work

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
