# Peer review of "Nanoparticles Assembled CdIn2O4 Spheres with High Sensing Properties towards n-Butanol"

_nanomaterials, 2019, doi:10.3390/nano9121714_

Round 1
Reviewer 1 Report
The manuscript describes hydrothermal synthesis of CdIn2O4 nanoparticles and its feasibility as a gas sensor for n –butanol detection. It is good written and might be interesting for the readership of Journal Nanomaterials. It can be accepted after major revision.
Here are my comments
1) Introduction part of manuscript should be modified.
2) Author could have examined the effect of synthesis reaction time on the growth process of CdIn2O4 The details of reaction growth mechanism is lacking. From material synthesis point of view there isn’t new protocol or chemistry.
3) Author should give more explanation on gas sensing mechanism. Fig. 8 (d-f) should be explained clearly and detail. The reasoning for the enhanced gas sensing performances should be elaborated or hypothesized. The schematic diagram can be helpful for the understanding.
4) How would you further improve the gas sensing properties of CdIn2O4 for butanol detection?
5) What is the effect of CdIn2O4 surface morphology on gas sensing performance? What are crucial factors? BET nitrogen sorption analysis can be done. EDS and elemental mapping can be done to provide more insights into structural features. After annealing at high temperature, author suggested material appears to be yellow, why?
6) The results should be compared with previous reported materials for butanol detection and can be compared with present work CdIn2O The summarized table including information on gas sensors can be added.
7) What is the state of art for such a systems? The recent advances in the current researches can be included.
8) What is the effect of mass activity or semiconductor amount on the gas sensor properties?
Author Response
Reviewer 1:
The manuscript describes hydrothermal synthesis of CdIn2O4 nanoparticles and its feasibility as a gas sensor for n -butanol detection. It is good written and might be interesting for the readership of Journal Nanomaterials. It can be accepted after major revision.
Here are my comments
1) Introduction part of manuscript should be modified.
Response: Thanks so much for your valuable suggestions. The introduction has been modified. For the purpose of clarity, we highlight these parts in yellow background.
2) Author could have examined the effect of synthesis reaction time on the growth process of CdIn2O4 nanoparticles. The details of reaction growth mechanism is lacking. From material synthesis point of view there isn’t new protocol or chemistry.
Response: In our experiment, we mainly focused on the effect of Cd/In ratio to obtain pure CdIn2O4 using glycerol as a coordinated reagent. Therefore, the time dependent experiment is excluded. In the revised manuscript, the reaction growth mechanism is proposed.
3) Author should give more explanation on gas sensing mechanism. Fig. 8 (d-f) should be explained clearly and detail. The reasoning for the enhanced gas sensing performances should be elaborated or hypothesized. The schematic diagram can be helpful for the understanding.
Response: As a resistive gas sensor, the main gas sensing mechanism is explained by the reaction between the target molecules and the absorbed oxygen species. Although deep understanding is highly expected, it is still unclear so far. Briefly, we combined schematic diagram and plausible reaction to give an explanation.
4) How would you further improve the gas sensing properties of CdIn2O4 for butanol detection?
Response: Several ways can be used to further improve the gas sensing properties of CdIn2O4 for butanol detection, as described in the revised version. For example, the general approaches include noble nanoparticles functionalization, creation of heterojunction by coupling with other sensing materials, improvement in specific surface area by introducing additives and templates in the reaction system and so on.
5) What is the effect of CdIn2O4 surface morphology on gas sensing performance? What are crucial factors? BET nitrogen sorption analysis can be done. EDS and elemental mapping can be done to provide more insights into structural features. After annealing at high temperature, author suggested material appears to be yellow, why?
Response: Generally, high surface area and porous structures will provide more active sites to favour the diffusion of gas molecules and reaction with oxygen species. In our case, the BET nitrogen sorption analysis is shown in Fig. 4, which reveals a small specific surface area of 0.68 m2/g with the pore sizes mostly around 30 nm. The relatively small specific surface area can be explained by the heave accumulation of CdIn2O4 nanoparticles. Besides, the EDS and elemental mapping have been provided in Fig. . It is can be seen that the Cd, In and O elements are homogenously distributed, which is quite close to the ration in the formula of CdIn2O4.
In the experiment, we described the change of product in color. After annealing at high temperature, the dried Cd-In precursor (greyish white in color) changed into yellow.
6) The results should be compared with previous reported materials for butanol detection and can be compared with present work CdIn2O4. The summarized table including information on gas sensors can be added.
Response: A comparison has been made in Table 1. Our CdIn2O4 based sensor exhibits comparable performance in response and recovery time and limit of detection.
7) What is the state of art for such a systems? The recent advances in the current researches can be included.
Response: We included more related researches in the revised manuscript to strengthen our background and discussion.
8) What is the effect of mass activity or semiconductor amount on the gas sensor properties?
Response: The alumina tube in our experiment is about 4 mm in length and 1.2 mm in external diameter and 0.8 mm in internal diameter. The thickness of the sensing materials is about 0.6-0.8 mm. Therefore, almost 1 mg sensing mateials have been loaded on it. As described in the experimental section, we estimate the response by the change of change in sensor resistance. Thus, the amount of the sensing materials is not a key factor. Thank you so much.

Reviewer 2 Report
The manuscript describes preparation and testing of a cadmium indate nanoparticle sensor for n-butanol. Starting material Cd/In molar ratio of 1:1.6 yielded pure phase CdIn2O4 and detection limit for butanol was about 1 ppm.
The introduction gives a good overview of the relevant sensor preparation and performance information. What is needed in the manuscript is a more comprehensive discussion in section 3 (which should be renamed Results and Discussion) of how results compare with previous work in the field. This is particularly important with respect to the statement on line 38 concerning synthesis of pure phase with controllable morphology. This seems to have already been accomplished in, for example, reference 18 at 1:1.8 ratio in aqueous medium to produce octahedral material. The achieved detection limit should also be discussed in relation to other sensor studies, such as the response of 6 reported in reference 12 for 1 ppm butanol concentration.
An explanation should be given in 2.1 as to why the synthesis method is called hydrothermal when the liquid phase used is glycerol/isopropanol.
The sensing mechanism starting on line 170 is general for hydrocarbons which can end up producing carbon dioxide and water on the sensor surface, equations 5 and 6. It thus gives no explanation to the selectivity for n-butanol shown in Figure 7 nor support for the statement on line 165 that the 280 C temperature somehow favors n-butanol. An extensive investigation of responses to the different vapors at different temperatures would be required to begin understanding the sensor selectivity. There does seem to be a trend for the n-alcohols in Figure 7 with responses increasing methanol<ethanol<butanol. Responses for n-propanol and n-pentanol would be interesting.
Heave accumulation on line 107 should read large accumulation or large concentration.
The x-axis in Figure 6f is in seconds but is given in days on line 155. It should also be made clear that the responses shown are during the continuous pulsing as in 6e and not constant exposure to the vapor concentrations shown.
"Decrease" on line 178 should be "increase". But increases and decreases of resistance are all relative to a reference condition, which according to line 75 is the sensor in air. It would be more logical to describe the sequence of events at the top of page 9 as the resistance decreases from that of the reference state upon exposure to butanol and then increases to that of the reference state when the butanol is removed.
Related to this is the somewhat confusing information presented in Figure 8. Figures a-c show as stated on line 183 that Cd, In and O are present in the sensor which is not surprising given the starting materials indicated in section 2. But the XPS measurements were done in vacuum and thus are not directly applicable to the reference condition in air. Figures d-f are a visual representation of equations 1-6. They give the impression of being a simulation of the sensing mechanism but have only artistic merit. It would be better to delete the figure.
Corrections should be made in the abstract: "Here," should be removed on line 10, "is" should be replaced with "was" on line 12, "by the Cd/In molar ratios" should be replaced with "for the Cd/In starting materials ratio" on line 13, "are assembled by a large number" should be replaced with" were composed" on line 13, "When using as sensing materials for n-butanol detection," should be removed on line 14 with new sentence starting with "The CdIn2O4", and "exhibits" should be replaced with "exhibited" on line 15.
Author Response
Reviewer 2:
The manuscript describes preparation and testing of a cadmium indate nanoparticle sensor for n-butanol. Starting material Cd/In molar ratio of 1:1.6 yielded pure phase CdIn2O4 and detection limit for butanol was about 1 ppm.
1) The introduction gives a good overview of the relevant sensor preparation and performance information. What is needed in the manuscript is a more comprehensive discussion in section 3 (which should be renamed Results and Discussion) of how results compare with previous work in the field. This is particularly important with respect to the statement on line 38 concerning synthesis of pure phase with controllable morphology. This seems to have already been accomplished in, for example, reference 18 at 1:1.8 ratio in aqueous medium to produce octahedral material. The achieved detection limit should also be discussed in relation to other sensor studies, such as the response of 6 reported in reference 12 for 1 ppm butanol concentration.
Response: Thanks so much for your comments. Based on the reviewers’ suggestions, we have modified our manuscript thoroughly. Although, some reports have provided the ratio for preparation pure CdIn2O4, a slight difference could occur because of the different synthesis routes.
3) An explanation should be given in 2.1 as to why the synthesis method is called hydrothermal when the liquid phase used is glycerol/isopropanol.
Response: The synthesis route has been changed into solvothermal method. Thank you.
4) The sensing mechanism starting on line 170 is general for hydrocarbons which can end up producing carbon dioxide and water on the sensor surface, equations 5 and 6. It thus gives no explanation to the selectivity for n-butanol shown in Figure 7 nor support for the statement on line 165 that the 280 C temperature somehow favors n-butanol. An extensive investigation of responses to the different vapors at different temperatures would be required to begin understanding the sensor selectivity. There does seem to be a trend for the n-alcohols in Figure 7 with responses increasing methanol<ethanol<butanol. Responses for n-propanol and n-pentanol would be interesting.
Response: For resistive gas sensor, the selectivity is mainly determined by the sensing mateirals and the working temperature because of the difference in active energy of different molecules. In our case, the operating temperature of 280  °C provides sufficient energies to favor the n-butanol gas molecule adsorb, desorb and react on the CdIn2O4 surface. At lower temperature, insufficient energies are not in favor of acting reaction between gas molecules and absorbed oxygen species. However, the absorbed gas molecules could greatly decrease at relative higher temperature.
5) Heave accumulation on line 107 should read large accumulation or large concentration.
Response: It has been changed. Thanks.
6) The x-axis in Figure 6f is in seconds but is given in days on line 155. It should also be made clear that the responses shown are during the continuous pulsing as in 6e and not constant exposure to the vapor concentrations shown.
Response: We apologize for the mistake. It has been changed into days in Fig. 6 (f). Besides, the response has been provided in Fig. 6 (e) to make it clear.
7) "Decrease" on line 178 should be "increase". But increases and decreases of resistance are all relative to a reference condition, which according to line 75 is the sensor in air. It would be more logical to describe the sequence of events at the top of page 9 as the resistance decreases from that of the reference state upon exposure to butanol and then increases to that of the reference state when the butanol is removed.
Response: Thanks so much. It has been corrected.
8) Related to this is the somewhat confusing information presented in Figure 8. Figures a-c show as stated on line 183 that Cd, In and O are present in the sensor which is not surprising given the starting materials indicated in section 2. But the XPS measurements were done in vacuum and thus are not directly applicable to the reference condition in air. Figures d-f are a visual representation of equations 1-6. They give the impression of being a simulation of the sensing mechanism but have only artistic merit. It would be better to delete the figure.
Response: Admittedly, the XPS spectra are recorded in vacuum. However, it reflects the oxygen species to some extent, which is widely used to discuss the gas sensing mechanism. In order to vividly show the sensing mechanism described by the equations 1-6, an schematic diagram is given in Fig. 8 (d-f).
9) Corrections should be made in the abstract: "Here," should be removed on line 10, "is" should be replaced with "was" on line 12, "by the Cd/In molar ratios" should be replaced with "for the Cd/In starting materials ratio" on line 13, "are assembled by a large number" should be replaced with" were composed" on line 13, "When using as sensing materials for n-butanol detection," should be removed on line 14 with new sentence starting with "The CdIn2O4", and "exhibits" should be replaced with "exhibited" on line 15.
Response: These changes have been made. Thanks very much for your valuable suggestions to improve our manuscript. I hope we addressed all your concerns related to our work.

Reviewer 3 Report
The manuscript entitled “Nanoparticles assembled CdIn2O4spheres with high sensing properties towards n-butanol” reports appreciable sensing results with good stability and reproducibility. In this current state the manuscript needs some characteristic validations and adequate correlations. Also, the article has some topographical and linguistic corrections to be processed. After the English and scientific correction, the paper can be accepted with major revision.
1. The paper on the whole gives very basic and simple analysis of n-butanol sensing. Thus, the paper lacks in novelty and hence authors must do additional analysis to improve the quality of the paper 2. Authors could report some limit of exposure or danger limit 3. Authors must explain the growth mechanism of the nanosphere since the technique is claimed to be a novel one 4. Authors could estimate the particle size distribution since the observed morphology is spherical 5. Authors could include the bandgap properties of the prepared nanostructures 6. Authors could give the pore size and surface area of all the prepared samples 7. TGA must be explained in detail 8. Effect of pore size and the action of pores in sensing mechanism could be correlated 9. From the dynamic response, 1 ppm does not show an appreciable response. Hence authors must conduct interference studies/real time analysis to validate the detection or must change the claimed detection limit 10. Comparison table could be included with room temperature butanol sensor 11. Selectivity of butanol is to be explained 12. It was found that the -OH group VOCs are responsive towards the prepared nanostructure; hence author must compare the butanol selectivity with other alcoholic groups 13. Lack of correlation in the manuscript 14. Authors might consider explaining the influence of Cd and In particles in the sensing mechanism 15. Authors might consider validating the sensing mechanism by checking CO2 emission testAuthor Response
Reviewer 3:
1) The paper on the whole gives very basic and simple analysis of n-butanol sensing. Thus, the paper lacks in novelty and hence authors must do additional analysis to improve the quality of the paper.
Response: Thanks a lot for your suggestion, which is great helpful for us to improve our manuscript. In the revised manuscript, the revised parts have been remarked in yellow background. Metal alkoxides appear a novel precursor-directed synthetic strategy to fabricate inorganic nanomaterials, in which the metal provides extra electrons to coordinate with the hydroxyl groups at alcohols to form stable compounds in solid. In our work, we firstly prepared Cd/In-glycerate by a solvothermal process. As a precursor, these Cd/In-glycerate was converted into CdIn2O4 after thermal treatment. Furthermore, the sensing properties for n-butanol detection are explored. Our work cannot only provide a way to prepare unique CdIn2O4 spheres, but also confirm its potential for n-butanol detection. According to the reviewer’s comments, the EDS and mapping analysis is further provided as well.
2) Authors could report some limit of exposure or danger limit
Response: These information related n-butanol has been given in the introduction. Thanks.
3) Authors must explain the growth mechanism of the nanosphere since the technique is claimed to be a novel one
Response: The growth mechanism has been proposed in the revised version. Please check it in the text.
4) Authors could estimate the particle size distribution since the observed morphology is spherical
Response: In Fig. 3 (c), the nanoparticles is determined to be dozens to more than one hundred nanometers from the broken CdIn2O4 spheres.
5) Authors could include the bandgap properties of the prepared nanostructures
Response: By surveying the literatures, the bandgap for CdIn2O4 is 3.2 eV, which is included in the introduction.
6) Authors could give the pore size and surface area of all the prepared samples
Response: Due to the different coordinated capacity of glycerol with Cd2+ and In3+ ions, we firstly discussed the effect of Cd/In ratio to the phase of final products. Then we mainly focused on the pure CdIn2O4.
7) TGA must be explained in detail
Response: It has been improved in the revised manuscript.
8) Effect of pore size and the action of pores in sensing mechanism could be correlated
Response: In general, porous nanomateriasl, mesoporous in particular, will favour the diffusion of gas molecules and provide more active sites to enhance the sensing performance.
9) From the dynamic response, 1 ppm does not show an appreciable response. Hence authors must conduct interference studies/real time analysis to validate the detection or must change the claimed detection limit
Response: Admittedly, interference gas molecules may affect the detection. Our CdIn2O4 based gas sensor exhibits response of 1.48 for 1 ppm n-butanol without interferences, which can reflect the detection limit to some extent, which is widely used to evaluate the sensing properties of a gas sensor.
10) Comparison table could be included with room temperature butanol sensor
Response: A comparison has been made in Table 1. The working temperature is important parameters to determine the sensing performance. So far, it is rarely found room temperature butanol sensor.
11) Selectivity of butanol is to be explained
Response: The good selectivity could be explained by the fact that the operating temperature of 280  °C provides sufficient energies to favor the n-butanol gas molecule adsorb, desorb and react on the CdIn2O4 surface. At lower temperature, insufficient energies are not in favor of acting reaction between gas molecules and absorbed oxygen species. However, the absorbed gas molecules could greatly decrease at relative higher temperature.
12) It was found that the -OH group VOCs are responsive towards the prepared nanostructure; hence author must compare the butanol selectivity with other alcoholic groups
Response: As shown in Fig. 7, the selectivity with other alcohols such as ethanol, methanol, isopropanol have been made. Indeed, the ethanol represents an obvious interference.
13) Lack of correlation in the manuscript
Response: The manuscript has been modified to enhance the correlation.
14) Authors might consider explaining the influence of Cd and In particles in the sensing mechanism
Response: As a resistive gas sensor, the main gas sensing mechanism is explained based on the reaction between the target molecules and the absorbed oxygen species. The roles of metal components in the sensing materials such as metal oxides, metal sulfides etc. are still unclear so far.
15) Authors might consider validating the sensing mechanism by checking CO2 emission test
Response: A trace of CO2 emission may be released during the test, thus it is really hard to check. Thank you so much. Hopefully, we addressed all your concerns.

Round 2
Reviewer 1 Report
Authors have made the possible corrections and modified the manuscript. I recommend the manuscript for acceptance in Journal Nanomaterials.
Reviewer 2 Report
Fig 5 "Relactive" on the x-axis should be "Relative"
line 118 "heave" should be "large"
line 260 "n-butanl" should be "n-butanol"
Author Response
Response: Thanks so much for reviewing our manuscript carefully. These mistakes have been corrected. We also check our manuscript again to avoid such mistakes.

Reviewer 3 Report
The quality of manuscript improved sufficiently after revision and it can be accepted for publication in Nanomaterials.
Author Response
Response: Thanks so much for your comments to greatly improve our manuscript.
